# Decoupling and Decomposition Analysis of Land Natural Capital Utilization and Economic Growth: A Case Study in Ningxia Hui Autonomous Region, China

**DOI:** 10.3390/ijerph18020646

**Published:** 2021-01-14

**Authors:** Shanshan Guo, Yinghong Wang, Jiu Huang, Jihong Dong, Jian Zhang

**Affiliations:** 1School of Public and Management, China University of Mining and Technology, Xuzhou 221116, China; guoshanshan@cumt.edu.cn (S.G.); LB20160018@cumt.edu.cn (J.Z.); 2School of Environment Science and Spatial Informatics, China University of Mining and Technology, Xuzhou 221116, China; jhuang@cumt.edu.cn (J.H.); dongjihong@cumt.edu.cn (J.D.)

**Keywords:** land, natural capital utilization, economic growth, decoupling analysis, decomposition

## Abstract

In order to reduce the depletion of land natural capital and develop economy simultaneously, it is necessary to study how to achieve the strong decoupling relationship between them. However, so far such studies have been relatively limited. Thus, taking the case of Ningxia Hui Autonomous Region, China, this paper firstly analyzes the state of land natural capital utilization in 1999–2017 by using improved ecological footprint. Then, decoupling state is quantified by Tapio decoupling model. Last, major driving factors on the decoupling relationship are explored with combination of LMDI decomposition and Kaya identity equation. Results showed that: (1) Both natural capital flows and stock depletion of cultivated land decrease obviously during the transition to corn-based intensive ecological agriculture. Grassland and water are the most unsustainable development sectors among all land types with their stock depletion intensified. Forest land and construction land could basically meet the consumer demand, but the flow occupancy of construction land is the fastest-growing segment. (2) Decoupling relationship is in an alternating state between weak decoupling and strong decoupling in 1999–2017. Wherein, the cultivated land and forest land showed a preferred decoupling state, followed by grassland, while the water and construction land showed the unfavorable expansive negative decoupling and weak decoupling. (3) Decomposition results show that intensity effect is the major factor that promotes the decoupling while economic effect inhibits the decoupling, but this negative impact is weakening in the process of industrial transformation. The other three factors affect less on the decoupling. This study has a certain reference value to construct an ecological civilization in eco-fragile regions and formulate relevant policies on the increase of land natural capital efficiency.

## 1. Introduction

With the significant population growth, urban expansion and the extensive use of powerful scientific and technological means, global depletion of natural resources has reached an unprecedented scale and become largely unsustainable [1,2]. This puts forward a severe challenge of the rational use of natural resource to increase the supply of ecosystem goods and services [3] and to maintain biodiversity that is vital to ecosystem functionality [4]. In this process, the Sustainable Development Goals (SDGs) proposed by the United Nations [5] has increasing been the focus of attention in the world, which enriches the insights of economists and policymakers into economic growth’s possible effects on the ecological environment, shifting their paradigms from the simple pursuit of economic growth to concern about ecologically friendly economic growth. The concept of ‘‘natural capital” originated from the idea of sustainable development, highlighting the efficient and sustainable use of natural resources [6]. Natural capital has replaced artificial capital as an important limiting factor for the well-being of human society today and natural capital management has increasingly become an important issue in the study of sustainable development [7,8,9]. Wherein, Land Natural Capital (LNC), as an important index and criterion, has played a significant role in achieving the effective management of resource and the economically, socially and environmentally sustainable development.

As the second-largest economy as well as most populous country in the world, China has been playing a key role in the process of global sustainable development and has formulated a set of policies to achieve this goal. However, the natural resource extraction and exploitation accelerate in parallel with economic growth that leads to decrease biocapacity while increasing the ecological footprint, which has posed a major barrier to China’s sustainable development [10,11]. In striving to promote environmental quality and resource efficiency, China has initiated the Ecological Civilization Construction [12], emphasizing the need to build a Life Community of Mountains, Rivers, Forests, Lakes and Grasses. This highlights the urgency for well-formulated metrics that could help policymakers better understand the progress on resource sustainable use regionally. But on the whole, there were few studies on qualify the relationship between economic development with ecological quality and environmental quality at home and abroad, and lacked of relevant researches on the utilization of resources in different stages of economic development.

Ningxia Hui Autonomous Region (Ningxia) is located primarily in arid and semi-arid zones, belonging to the typical ecologically fragile region and one of the most desertified provinces in China. Beset by drought and lack of water resources as well as unreasonable resource utilization and long-term overgrazing and overcultivation, the fragile ecological environment is facing greater challenges, seriously restricting the sustainable development of society and economy in Ningxia. In order to better develop the pilot area of ecological civilization construction and industrial transformation in western region of China, National Development and Reform Commission (NDRC) of Ningxia issued the Comprehensive Management Plan for Key Ecological Areas in Western Region (2012–2020) and Action Plan of Boosting the Development and Transition of the Western Region (2013–2020) [13], highlighting that Ningxia should undertake important task of ecological management and ecological construction in the western region of China. Therefore, as a typical representative of the western provinces in both fragile environment and economic transformation, research on the characteristics of land natural capital utilization (LNCU), decoupling economic outputs from land occupancy, as well as the primary factors affecting their decoupling nexus, is conducive to achieving the goal of accelerating natural capital appreciation meanwhile providing guidance to the economic transition in ecologically fragile regions.

## 2. Literature Review

At present, various measurement and accounting methods of natural capital utilization have been explored by scholars in studies on sustainable development goals monitoring. Of the latter approaches, the ecological footprint (EF) is defined as a bio-productive land area that maintains human living needs while absorbing pollution caused by human activities [14] from the ecological perspective, which has been broadly used to reflect environmental degradation because it focuses on production and consumption activities on the environment both directly and indirectly [8]. The outstanding merits of this method are that it is easy to apply, repeatable, and simple to understand [15,16]. To date, the ecological footprint model has been improved and extended from one-dimension [17] to two-dimension [14,15] and then to three-dimensional model [18,19], and has been applied to researches at multiple scales of global [15], national [20], province [21], city [22], and enterprise [23]. Especially in ecologically sensitive areas [24,25], the ecological footprint model has fully demonstrated its advantages of ecological sustainability. In the latest applications of three-dimensional ecological footprint in Shandong Province [26], Jiangsu Province [27], Hainan Province [28], Shaanxi Province [29], Beijing-Tianjin-Hebei Metropolitan region [30], Pearl River Delta Urban Agglomerations [31], Yellow River Delta Region [32], Southern Qin Ling piedmont [33], Beijing city [34] and Guiyang city [8], scholars studied the dynamics of regional natural capital stock and flow, ecological sustainability and driving factors influencing the changes of ecological footprints. While this metric is appealing as a communication tool for showing human impact on the natural resources and environment, its methodology and usefulness have also been challenged. Previous researches mostly involve the ‘‘real state” ecological footprint (land footprint) and ‘‘virtual state” ecological footprint (energy footprint) into natural capital accounting for the purpose of incorporation of the key natural capitals that underpin human society as fully as possible. But the point is that energy footprints do not exist actual ecological capacity correspondingly [9], which results in an irrationally high of land footprint depth. Therefore, Fang [8] discussed defining the “real state” land ecological footprint from the perspective of production to account for the dynamic changes of natural capital utilization of urban and rural land in Guiyang city based on the improved three-dimensional ecological footprint. Above all, despite intense research based on the three-dimensional ecological footprint, there are less researches on the LNCU and its balanced relationship with regional economic in terms of ‘‘real state” land footprints.

The core process of human-land relationship is mainly embodied in the relationship between the pressure of human activities and the pressure capacity of resources and environment [35]. At present, various theoretical approaches, such as bivariate causality, correlation analysis, multivariate cointegration, simple regressions, variance decomposition, have widely applied to research the causal relationship between land resource utilization and economic growth [36,37,38]. However, whether an economy is becoming less dependent on land capital consumption has become another important issue that was less on information. The decoupling analysis has become an important method to study that problem. This theory, originated from the concept of physics, was originally applied to explain the decrease or absence of interrelationships between physical quantities [39]. Later, the Organization for Economic Cooperation and Development (OECD) [40] firstly introduced the theory to analysis the fields of resources and environment in the early 21st century. However, the decoupling index defined by OECD varies with the selection of base period and is classified decoupling as relative or absolute, which cannot accurately reflect the decoupling nexus between those two indicators. To deal with this problem, Tapio [41] defined the decoupling index based on the idea of elastic coefficient and further divided the decoupling state into eight logical possibilities, i.e., strong decoupling (SD), weak decoupling (WD), recessive decoupling (RD), expansive coupling (EC), recessive coupling (RC), expansive negative decoupling (END), weak negative decoupling (WND) and strong negative decoupling (SND). Then numerous studies are available in analyzing the decoupling nexus between economic growth and energy consumption [42,43], economic growth and urban expansion [44] as well as GDP and environmental and resource interrelationships [45,46]. For example, Wang et al. [47] compared both the carbon emissions and the decoupling performance between China and the United States from 2000 to 2014, and found that China has experienced expansive coupling and weak decoupling, while the U.S. experienced mostly weak and strong decoupling. Song [48] examined the decoupling effects of economic growth on cultivated land decreases in China, and found that cultivated land was separately occupied by the expansion of urban areas (1986–1995), rural settlements (1995–2000) and “other construction” land use category (2000–2005) at different stages of China’s economic development.

Nevertheless, decoupling indicator method alone cannot evaluate the effects of environmental externalities and obtain the genuine feedback for improvement [49]. To overcome this shortcoming, another group of scholars have concentrated their efforts on investigating the inner mechanism of decoupling by integrating decoupling index with the decomposition approach. At present, the Structural Decomposition Analysis (SDA), the Index Decomposition Analysis (IDA), and the Production-theoretical Decomposition Analysis (PDA) are the three primary types of decomposition approaches [47]. Among which, the Logarithmic Mean Division Index method (LMDI), as the most representative of the IDA decomposition method, performs best because of the advantage of decomposition without residuals [50]. Therefore, it is regarded as the most accurate and practical method in the current decomposition system [51]. Zhang et al. [10] studied the decoupling relationship between coal consumption and economic growth in China, uncovering that the energy intensity effect played the crucial role in decreasing coal consumption, while the economic activity effect and population effect propelled the continual increase of coal consumption in 1991–2013. Liu et al. [52] adopted the LMDI method to decompose the changes of rural residential land area into rural residential land intensive use effect, urban-rural population structure effect, urbanization effect and total population effect. They found that the urban-rural population structure effect was the strongest driver in China’s rural residential land changes.

The preceding literatures lay important theoretical foundations and provide feasible and effective methods to study the situation of natural capital utilization as well as the decoupling relationship between resource depletion and economic growth. However, there is little research on the LNCU and its nexus with economic growth. The contributions of this article are focused on the following three aspects: (1) The status of LNCU is analyzed based on the improved EF method in Ningxia during 1999–2017; (2) The Tapio model is introduced to examine the decoupling nexus between LNCU and economic growth; (3) Instead of analyzing the influencing factors of LNCU, this paper systematically studies the major factors affecting the decoupling relationship with combination of the Logarithmic Mean Divisia Index (LMDI) method and Kaya identity equation.

## 3. Methods and Data Sources

### 3.1. Study Area and Data Resource

The Ningxia Hui Autonomous Region (104°17′–107°39′ E, 35°14′–149 39°23′ N) is located at the middle and upper reaches of the Yellow River, and the transition zone between Loess Plateau and desert in western China, belonging to the typical ecologically fragile area in western China, with the average annual precipitation and evaporation about 305 mm and 1800 mm separately. Therefore, most area of the region is now in a state of extreme drought and prone to a variety of natural disasters. The provincial area is 66,400 km^2^, including five prefectures (Figure 1). By 2017, the area of cropland, grassland and forest land respectively accounted for 19.5%, 31.4% and 11.6% of the total area [53]. The grassland is the dominate land use type. The per capita GDP of Ningxia was 5.1 × 10^4^ RMB in 2017, which only accounted for 85% of China’s average per capita GDP [54], facing the double pressure of environmental protection and economic growth. In recent years, Ningxia is undergoing the critical period of economic transformation and territorial space planning. Easing the pressure on the environment, reducing economic dependence on land resource and improving the efficiency of resource utilization are of great importance to achieve environmental protection and economic transition in Ningxia Province.

In this paper, considering that the time-series data in Ningxia started in 1999 and the latest data available to us was 2018, we selected the research period from 1999 to 2017 to avoid inaccuracies caused by inconsistent statistical coverage before 1999. The biological resource production data and socioeconomic data used herein are collected from China Statistical Yearbook (2000–2018) and Ningxia Statistical Yearbook (2000–2018). Wherein, the GDP data take the constant price in 2000 as the base year, and further obtained the actual GDP value. Additionally, we introduced the Liu’s [55] research on yield factors: the cultivated land and construction land are 0.94, the grassland is 2.25, the forest land is 0.85 and the water area is 2.25. The equivalence factor is adopted from the National Ecological Footprint Accounting Guidance: the cultivated and construction land are 2.52, the grassland is 0.46, the forest land is 1.29, and the water area is 0.37.

### 3.2. Improved Ecological Footprint Method

In this paper, we applied Fang’s [8] improved ecological footprint method to calculate the “real state” land EF from a productive perspective, that is, the land EF is defined as the area of biologically productive land (cropland, forest land, grassland, water and construction land) needed to support the production of regional biological products (agricultural products, forest products, livestock products and aquatic products) and the expansion of construction land [9,56]. The corresponding land capacity is defined as the area of biologically productive land actually available in a region. The calculation formulas for land footprint and land ecological capacity are shown below:(1)$$LF=\sum_{j=1}^{n}\sum_{i=1}^{m}\left(\frac{P_{ij}}{AP_{w,ij}}\times r_{j}\right)$$
(2)$$LC=0.88\times \sum_{j=1}^{n}\left(A_{j}\times r_{j}\times y_{j}\right)$$
where *LF* represents the total land footprint (gha) in a certain year, and *LC* indicates the total land ecological capacity(gha). $P_{ij}$ is the amount of product *i* harvested from the land use type *j* (t yr^−1^), $AP_{w,ij}$ is the global average yield of product *i* in land use type *j* (t wha^−1^ yr^−1^), $A_{j}$ represents the area of the biologically productive land required, $r_{j}$ and $y_{j}$ are the equivalence factor and the yield factor for a given land use type, respectively, *n* and *m* are the number of land use types and product items, respectively. According to the requirements of the World Commission on Environment and Development (WCED), the area of biologically productive land is required to reduce by 12% for biodiversity conservation [57].

The $LF_{siz,reg}$ is defined as the annual occupation area of bio-productive land within the limit of land capacity, which can represent the scale of human utilization of natural capital flow [8,18]. $LF_{dep,reg}$ is the portion of land natural capital occupation that exceeds the range of the land capacity. The formulas are as follows:(3)$$LF_{siz,reg}=\sum_{j=1}^{n}\min\left\{\left.LF_{j},LC_{j}\right\}\right.$$
(4)$$LF_{dep,reg}=1+\frac{\sum_{j=1}^{n}\max\left\{\left.LF_{j}-LC_{j},0\right\}\right.}{\sum_{j=1}^{n}LC_{j}}$$
where $LF_{j}$ and $LC_{j}$ represent the land footprint and land ecological capacity for the *j*th land use type (gha), separately. $LF_{siz,reg}$ is the regional land footprint size of natural capital flow (gha), and $LF_{dep,reg}$ is the regional land footprint depth of natural capital stock (gha).

When the land capital flow of a certain region is not completely occupied, the $LF_{dep,reg}$ equals 1. Utilization efficiency of capital flows ($UE_{flo}$) is introduced to represent the actual efficiency of human utilization of land capital flow. While when the land capital flow is completely occupied, the use ratio of land capital stocks to flows ($UR_{flo}^{sto}$) is introduced to represent the extent to which the land capital stock exceeds the capital flow, following the formulas:(5)$$UE_{flo}=\frac{LF_{size}}{LC}\times 100\%\;(LF\leq LC)$$
(6)$$UR_{flo}^{sto}=\frac{LF-LF_{size}}{LF_{size}}=LF_{size}-1(LF>LC)$$

### 3.3. Decoupling Indicator

Tapio decoupling theory breaks the linkage of resource and environmental pressure from economic benefits [41] by comparing the change rates between variables. Therefore, based on this theory, the decoupling elasticity index between LNCU and economic growth can be illustrated as:(7)$$\beta_{t}=\frac{\delta LF_{t}}{\delta G_{t}}=\frac{\Delta LF}{LF}/\frac{\Delta G}{G}=\Delta LF\times G/LF\times \Delta G$$
where $\beta_{t}$ denotes decoupling elasticity index, $\delta LF_{t}$ and $\delta G_{t}$ represent the growth rate of land capital consumption and gross domestic product (GDP) from a base year 0 to a target year *t*, respectively. ΔLF and ΔG represent the change value of land capital utilization and economic growth.

Table 1 presents eight grades corresponding to $\beta_{t}$. The growth rate of GDP and the indicator of *LF* can be coupled, decoupled or negatively decoupled [41,58]. Strong decoupling means that the land natural capital consumption declines while GDP grows. And it’s the preferred state for development. Achieving a strong decoupling is also the key point in double intensive of LNCU and economic growth. On the contrary, the strong negative decoupling is the most unfavorable state for the economic development, in which the land natural capital consumption increases and GDP declines. Coupling donates that there is synchronization between these two.

### 3.4. Decomposition Model

Since the Tapio decoupling model merely considers the decoupling relationship between LNCU and GDP, but does not explore the reasons behind the changes. Therefore, in this paper, several influencing factors of LNCU is decomposed by combination of Kaya identity, LMDI decomposition and Tapio decoupling model. Based on extended Kaya identity [59], the influencing factors can be divided into five parts: structure effect, technical effect, economic effect, labor force effect and population effect, which are defined as follows:(8)$$FL_{t}=\sum_{i=1}^{5}LF_{it}=\sum_{i=1}^{5}\frac{LF_{it}}{LF_{t}}\times \frac{LF_{t}}{Y_{t}}\times \frac{Y_{t}}{L_{t}}\times \frac{L_{t}}{P_{t}}\times P_{t}=\sum_{ij}S_{it}^{LF}\times I_{t}^{LF}\times M_{t}^{LF}\times N_{t}^{LF}\times P_{t}^{LF}$$
where $LF_{it}$ represents the footprint of the *i*th land use type in year *t*. *Y_t_*, *L_t_* and *P_t_* are the gross domestic product (GDP), the number of working population and resident population respectively in year *t*. $S_{it}^{LF}$ is the share of the *i*th land use type footprint to the total land footprint in year *t*, which represents structure effect; $I_{t}^{LF}$ is the land footprint depletion caused by ten thousand yuan GDP in year *t*, which represents technical effect, $M_{t}^{LF}$ is the output per unit of labor in year *t*, which represents economic effect; $N_{t}^{LF}$ is the size of labor force in year *t*, which represents labor force effect; $P_{t}^{LF}$ is the number of resident population in year *t*, which represents population effect.

According to LMDI decomposition and Kaya identity, the change of *LF* from a base year 0 and a target year *t*, represented by ΔLF, can be decomposed into five effects as follows: (i) the changes in the structure effect (represented by $\Delta LF_{S}$); (ii) the changes in the intensity effect (represented by $\Delta LF_{I}$); (iii) the changes in the economic effect (represented by $\Delta LF_{M}$); (iv) the change in the labor force effect (represented by $\Delta LF_{N}$), and (v) the changes in the population effect (represented by $\Delta LF_{P}$), as shown in Equation (9):(9)$$\Delta LF=LF_{t}-LF_{0}=LF_{S}+LF_{I}+LF_{M}+LF_{N}+LF_{P}$$

Each effect in the right-hand side of Equation (9) can be expressed as follows:$$\begin{array}{ll} \Delta LF_{S}=\sum_{i=1}^{5}\frac{LF_{it}-LF_{i0}}{\ln(LF_{it})-\ln(LF_{i0})}\times \ln(\frac{S_{it}^{LF}}{S_{i0}^{LF}}) & \Delta \;LF_{I}=\sum_{i=1}^{5}\frac{LF_{it}-LF_{i0}}{\ln(LF_{it})-\ln(LF_{i0})}\times \ln(\frac{S_{t}^{LF}}{S_{0}^{LF}})\\ \Delta LF_{M}=\sum_{i=1}^{5}\frac{LF_{it}-LF_{i0}}{\ln(LF_{it})-\ln(LF_{i0})}\times \ln(\frac{M_{t}^{LF}}{M_{0}^{LF}}) & \Delta LF_{N}=\sum_{i=1}^{5}\frac{LF_{it}-LF_{i0}}{\ln(LF_{it})-\ln(LF_{i0})}\times \ln(\frac{N_{t}^{LF}}{N_{0}^{LF}})\\ \Delta LF_{P}=\sum_{i=1}^{5}\frac{LF_{it}-LF_{i0}}{\ln(LF_{it})-\ln(LF_{i0})}\times \ln(\frac{P_{t}^{LF}}{P_{0}^{LF}}) & \end{array}$$

Combining Equations (7) and (9), a decoupling model between *LF* and GDP is established based on the LMDI method, which can decompose the decoupling elasticity index for the relationship between LNCU and GDP into corresponding decoupling elasticity indexes.
(10)$$\begin{array}{ll} \beta_{t} & =\frac{\Delta LF}{LF}/\frac{\Delta G}{G}=\Delta LF\times G/LF\times \Delta G\\ & =(\Delta LF_{S}+\Delta LF_{I}+\Delta LF_{M}+\Delta LF_{N}+\Delta LF_{P})\times G/LF\times \Delta G\\ & =\beta_{S}+\beta_{I}+\beta_{M}+\beta_{N}+\beta_{P} \end{array}$$

## 4. Results and Discussion

### 4.1. Analysis of the Land Natural Capital Utilization

Based on the method illustrated in Section 3.2, the land footprint and land capacity, land footprint size and land footprint depth, as well as the land footprint utilization efficiency of land capital flows and the use ratio of land capital stocks to flows can be obtained.

#### 4.1.1. Natural Capital Utilization of Cultivated Land

Cultivated land is the main component of Ningxia’s ecological footprint. As shown in Figure 2a, the cultivated land footprint increased rapidly from 1999 to 2012, and then leveled off during the following years at around 4.13 gha. Specifically, the footprints of rice and wheat showed a significant downward trend, and their proportions of total land footprint decreased by 23.23% and 17.02% to 0.092 gha and 0.505 gha respectively over the past 19 years. Oppositely, the proportion of corn increased obviously to 77.02%, followed by oil plants and vegetables (Figure 2b). This shows that Ningxia is constantly optimizing the agricultural planting structure to realize the transformation from extensive to intensive, decentralized to large-scale specialized modern agriculture. The footprint depth increased first and then decreased significantly in 2013, resulting in the large reduction of $UR_{flo}^{sto}$ (Figure 3). But the values of footprint depth were greater than 1, which meant the capital stock consumption of cultivated land was still relatively serious, and food supply pressure would continue to exist with the decrease of cultivated land per capita.

#### 4.1.2. Natural Capital Utilization of Grassland

The livestock products produced from grassland provide a large amount of animal protein and are one of the most important sources of food consumption for Ningxia residents. Figure 4a showed that the grassland footprint increased from 0.321 gha in 1999 to 0.803 gha in 2017, showing an increase of 1.50 times. Specifically, the footprint of pork decreased from 10.05% in 1999 to 8.11% in 2017, while footprints of beef, mutton and milk increased obviously from 7.65%, 6.48% and 3.45% to 22.29%, 20.24% and 21.52% separately (Figure 4b), showing the changes in people’s diet structure. In 1999, the grassland footprint depth was 3.247. By 2017, the footprint depth reached 12.699, suggesting that almost 13 times the current grassland area was required to support the livestock consumption in Ningxia and that this situation was likely to continue well into the future as urban and rural residents demanding more meat, eggs and dairy product. Based on the model hypothesis, the land capital flow is assumed to be used first, followed by capital stock [8]. However, during the process of using actual land natural capital, the footprint size (capacity) of grassland was rather low and had fallen by 56.28% since 1999, which exacerbated the contradiction between supply and demand of grassland resources.

#### 4.1.3. Natural Capital Utilization of Forest Land

Affected by human activity, the forest land footprint in Ningxia fluctuated more dramatically. From 1999 to 2006, forest land footprint firstly showed a mild downward trend but was generally sustained at around 0.311 gha, then it showed a rapid increase during 2007–2009 and peaked at 0.379 gha in 2007, and finally it declined rapidly to 0.101 gha in 2017 (Figure 5a). Forest land capacity changed similarly with footprint size from 1999 to 2015. But since then, forest land capacity remained stable at around 0.118 gha while the footprint size declined again for the reason that Ningxia enforced effective measures like Grain for Green Project and Three North Shelterbelt Forest Program to strictly control ecological land occupation during the early stage of the study period because of serious soil erosion. However, as the accelerating urbanization and implementation of the strictest farmland protection system, construction land occupied a part of forest land. Therefore, footprint depth of forest land showed a rapid volatility drop until it leveled off in 2015 at 1, meaning existing forest land could basically meet the current consumption demand.

It is noted that if the energy footprint was incorporated into the forestland to calculate the footprint depth according to the practice of Niccolucci [19], the forestland footprint depth of Ningxia in 2017 was up to 12.69 [60], which is obviously too pessimistic to reflect the actual situation of the local forest ecosystem.

#### 4.1.4. Natural Capital Utilization of Water

The water footprint in Ningxia increased from 0.082 gha in 1999 to 0.339 gha in 2017, up more than tripled (Figure 5b). The water capacity (footprint size) was slowly increasing year by year at an annual rate of 1.89%, from 0.004 gha in 1999 to 0.006 gha in 2017. The explosive growth of water deficit has not met the people’s greater demand of aquatic products. The water footprint depth experienced a substantial upswing with a change process of “N” shape over the past 19 years, resulting in the maximum level of $UR_{flo}^{sto}$ in five land use types. Then, the water footprint depth soared again from 28.337 in 2009 to 54.578 in 2017 after a brief decline in 2009, much higher than the water footprint in eastern China (2.67 in Zhejiang Province) [61], suggesting that water is the most unsustainable development sector among all land types and almost 55 times the current area is needed to sustain the water resource consumption. Moreover, as a severe shortage of water resources and the water capacity that can be provided in Ningxia is observed, the imbalance between water supply and demand will continue to deteriorate under the existing consumption patterns. Therefore, it is the fundamental way to adjust the aquaculture structure and increase the output per unit of aquatic product while protecting the lake and wetland.

#### 4.1.5. Natural Capital Utilization of Construction Land

The construction land includes urban land, industrial and mining land, transportation land and other built-up land types, which is most closely related to people’s daily production and life. As showed in Figure 6a, construction land footprint (footprint size) increased rapidly from 0.006 gha in 1999 to 0.023 gha in 2017, representing a 7.59-fold increase. Although construction land occupied the smallest area of biological productive lands in Ningxia, it was the fastest-growing segment. The construction land capacity remained relatively stable in 1999–2008, and then began to increase after a slight drop in 2009, suggesting that with the expansion of urban scale in Ningxia, urban construction land use pattern is gradually optimized. Over the past 19 years, the footprint depth maintained at 1, and the utilization efficiency of capital flows increased from 4.639% to 21.228%, with an average annual growth rate of 8.82% (Figure 6b), showing that there was still a large ecological surplus space and the land capital flow could meet the demand of urbanization development.

### 4.2. Analysis of the Decoupling Relationship

Based on the method illustrated in Section 3.3, the decoupling status between economic growth and LNCU during 1999–2017 can be easily verified, listed in Table 2. Overall, as the growth rate of economy was positive throughout, only three statuses occurred in the eight decoupling states, that is, expansive negative decoupling, weak decoupling and strong decoupling.

#### 4.2.1. Decoupling State of Economic Growth and Land Capital Utilization

In Table 2, there was a strong decoupling in 1999–2000, which is the optimal decoupling state, indicating the land natural capital occupation declined while the economic growth increased. But soon after, decoupling state gradually deteriorated and showed the weak decoupling and expansive coupling in 2000–2002, which meant that the growth trend of land capital consumption was almost the same as that of economy. During the year of 2003–2010, such decoupling state had improved: economy kept stable growth, while land capital consumption maintained slow growth, and even showed negative growth in 2006–2007. Therefore, this period was basically stable in the state of weak decoupling except for the strong decoupling in 2006–2007. While, such relatively stable state was broken in 2000–2014 due to the changeable rate of land capital consumption, and there became alternative between weak decoupling and strong decoupling. Since then, decoupling state between land capital utilization and economic growth has been remained preferred strong decoupling, suggesting that marginal effect of economic growth is smaller than that of land natural capital utilization.

#### 4.2.2. Decoupling State of Economic Growth and Each Bioproductive Land Type

Table 3 summarizes the decoupling status between five bioproductive land types and economic growth. Overall, both cultivated land and forest land showed strong decoupling from economic growth, while grassland, water and construction land experienced weak decoupling from economic growth. These clear periodical characteristics illustrated the differences in the priorities of regional development policies in different periods.

Specifically, the decoupling statuses between cultivated land capital use and economic growth experienced a positive process from expansive coupling to strong decoupling. In 1999–2003, the decoupling status fluctuated a lot and showed an obvious circleprocess of SD-WD-RD-SD. And then, there was a long period of weak decoupling in 2003–2010 except for 2006–2007 and 2010–2011 in which it showed the preferred strong decoupling. With continuous decrease of cultivated land footprint during the period 2012–2017, the decoupling relationship at this stage was generally stable at strong decoupling except for weak decoupling in 2013–2014. Due to strict ecological protection measures, natural capital consumption of forest land continued to decrease and showed the optimal decoupling state. In 1999–2017, the decoupling index of forest land maintained a preferred strong decoupling except for weak decoupling in 2004–2005 and 2008–2009, as well as expansive negative decoupling in 2006–2007.

During the year of 1999–2001, the decoupling index of grassland capital use and GDP growth showed expansive negative decoupling. In the following period this situation got better and appeared a long weak decoupling, even some strong decoupling status occurred intermittently in 2005–2006 and 2010–2011. Decoupling statuses of water capital use and GDP growth experienced a deteriorated trend from weak decoupling in 1999–2000 to expansive coupling in 2000–2001, and further to expansive negative decoupling in 2001–2002. Then, this situation improved and appeared a seven-year weak decoupling in 2002–2006 and 2007–2010. However, during the year of 2010–2014 the decoupling state deteriorated again and showed the unfavorable expansive negative decoupling throughout. Until recent years, the growth rate of water footprint slowed down significantly and decoupling state returned back to weak decoupling. Compared with other decoupling indexes of productive land types, the decoupling state between construction land and economic growth was poor and fluctuated a lot. In 1999–2007, the decoupling status of construction land undergone the transition from the strong decoupling to the expansive negative decoupling. Then after a brief preferred decoupling in 2007–2009, it returned to expansive negative decoupling again in 2010–2011. While, during the following years, the decoupling state stepwise improved to strong decoupling along with construction land control and spatial layout optimization.

### 4.3. Analysis of the Driving Factors on Decoupling Relationship

According to the decomposition Equation (10) presented in Section 3.4, land natural capital utilization was jointly determined by five main factors. Figure 7 depicts the decomposition results listed in Table 4 to illustrate the contribution of each index to the decoupling state in every year from 1999–2017. As the growth rate of GDP is positive throughout, the negative value of decoupling index indicates a promoting effect to the decoupling between LNCU and economic growth. On the contrary, it performs a negative effect on the decoupling state.

#### 4.3.1. Structural Effect

Figure 7 presents the impact of structural effect changes on decoupling for the period from 1999 to 2017. Overall, the impact of this factor was small and in slight fluctuations. However, this negative impact had weakened with respect to the zero point of decoupling index, and even appeared positive impacts on the decoupling in the year 1999–2000, 2002–2003 and 2012–2013 due mostly to the large reduction of natural capital consumption of cultivated land. It indicated that the adjustment and optimization of land use structure could play a role in decoupling economic growth and LNCU to a certain degree.

#### 4.3.2. Intensity Effect

The intensity effect reflects the impact of technological progress on the decoupling. As shown in Table 4, the decoupling index of intensity effect was significantly less than 0 throughout but fluctuated a lot, indicating that the improvement of technology played a dominant role in decoupling. In recent years, with establishment of Ecological Agriculture Demonstration Area, the High-tech Development Zone and Inland Opening-up Pilot Economic Zone in Ningxia, scientific and technological progress have greatly improved regional capital utilization efficiency and supply capacity. While, during the year of 2013–2017, this positive impact decreased a bit. For instance, the intensity effect weakened and its decoupling index showed a downward trend due to the marginal utility (Figure 7).

#### 4.3.3. Economic Effect

The results presented that the decoupling indexes of economic output were positive throughout, with a mean value of 0.687, which was much greater than that of structural effect, labor force effect and population effect. It means that economic profits in Ningxia mainly rely on the consumption of natural resource and result in the most obvious negative impact on the decoupling. As shown in Figure 7, the curve of economic effect can be divided into three stages: a rapid-increase stage from 1999 to 2007, a great fluctuation period from 2009 to 2011, and a steadily decrease phase from 2012 to 2017, reaching the minimum 0.236 in 2014–2015. This change can explain the transition of economic pattern from resource-dependent development to innovation-independent economy in Ningxia.

#### 4.3.4. Effects of Labor Force and Population

It can be observed from Table 4 that during 1999–2017, the effects of labor force and population showed an opposite change characteristic and a negative impact on the decoupling as a whole. Specially, population effect fluctuated marginally around its mean value of 0.099 during the research period, indicating that with the continuous development of the economy of Ningxia, the status quo of the populous province has not been significantly changed. On the contrary, the labor force appeared a rapid fluctuation in the year of 2007–2011. The reason is that due to economic slowdown in Ningxia, large numbers of labor forces in the secondary and tertiary industries have shifted to the agricultural labor force. During this period, the decoupling index between GDP growth and LNCU increased rapidly. Nevertheless, from 2011 onwards the influence of this factor on the decoupling has weakened and gradually stabilized around 0.1 owing to the concentration of urban population in the accelerating process of industrialization and urbanization.

## 5. Conclusions and Policy Implications

Natural capital utilization in typical ecologically sensitive and fragile areas has attracted wide attention. Although existing studies have contributed theoretical foundations and various methods to study the dynamic change of natural capital utilization from multiple scales, the ‘‘real state” land footprint analysis of LNCU research is still insufficient and there are less studies concentrated on the decoupling relationship between natural capital depletion and economic growth. Therefore, in this paper, we established the improved EF model to analysis the states of LNCU in Ningxia Province, introduced Tapio decoupling theory was to explore the nexus between LNCU and economic growth, and decomposed the major factors affecting this relationship in combination with the Kaya identity and LMDI model, which further expands the existing research. However, despite our promising results, there are still some limitations in this study: first of all, we mainly focused on the perspective of production to study the changes in the use of land natural capital, which may produce different results from the perspective of consumption. Second, our study adopts a top-down approach, and most data used are from the Ningxia Statistical Yearbook and the China Statistical Yearbook. When calculating the ecological footprint of the city and county scale, the accuracy of the data is insufficient, and therefore the further research on a small scale should be carried out in combination with field investigation. The primary conclusions are as follows:(1)From the analysis of land natural capital utilization in Ningxia, it can be observed that the natural capital stock utilization of cultivated land decreased obviously, resulting in the declining trend of the $UR_{flo}^{sto}$
in recent years. Similarly, in forest land, it decreased constantly and the flow occupation of natural capital could basically meet consumer demand since 2015. While, the $UR_{flo}^{sto}$
of grassland and water increased rapidly and performed as the most unsustainable sectors among all land types. Moreover, the footprint of construction land occupied the smallest area of biological productive lands in Ningxia, but was the fastest-growing segment.(2)The decoupling analysis showed that the pressure of economic development on the sustainable use of land natural capital always exists. Overall, the decoupling state was preferred and dominated by strong decoupling and weak decoupling. The two stages appeared almost at the same frequency and only in 2001–2002 there was an expansive coupling. Specially, the cultivated land and forest land showed a preferred decoupling state in recent years, followed by grassland, while water and construction land showed the unfavorable expansive negative decoupling and weak decoupling.(3)In terms of the decomposition results, it can be obtained that economic effect is the biggest unfavorable factor limiting the strong decoupling, while intensity effect is the most favorable factor to promote the decoupling of land capital occupation from economic growth in Ningxia Province. Moreover, the impacts of structural effect, labor force effect and population effect on the decoupling are relatively weak.

Given the above findings and regional characteristics of Ningxia, we propose the following policies to promote the rational use of land natural capital and the decoupling of land capital occupation from economic growth in Ningxia based on the partition perspective:(1)In the central and southern mountains with fragile ecological environment (Guyuan city, South of Wuzhong City and Zhongwei city), considering the reality that grassland resource is abundant but overutilized seriously, measures related to the grassland and forest land protection, such as the Grain for Green Project, the Region-Wide Grazing Ban and the Three North Shelterbelt Forest Program, should be strictly implemented to improve ecological carrying capacity. Apart from that, due to the rapid expansion of urban construction land, later monitoring and management are crucial for the maintenance of ecological restoration achievements.(2)In the northern Yellow River irrigation area where is featured with higher economic development and better ecological environment (Yinchuan city and Shizuishan city), it is imperative to promote the transformation and upgrading of traditional industries and advance scientific and technological innovation for the purpose of reducing the impediment of economic effect on the decoupling relationship, and improving the intensity effect by improving the efficiency of resource utilization. At the same time, reasonably controlling the boundary of urban space expansion and optimizing the population structure to give full play to the powerful driving effect of talents on the economy instead of the overuse of natural resources.(3)Moreover, in the Ningxia Plain along the Yellow River, it is suggested to vigorously develop ecological, water-saving agriculture, reduce the use of fertilizer and control the loss of water resources, to alleviate the pressure of the extreme shortage of water resources.

## Figures and Tables

**Figure 1 ijerph-18-00646-f001:**
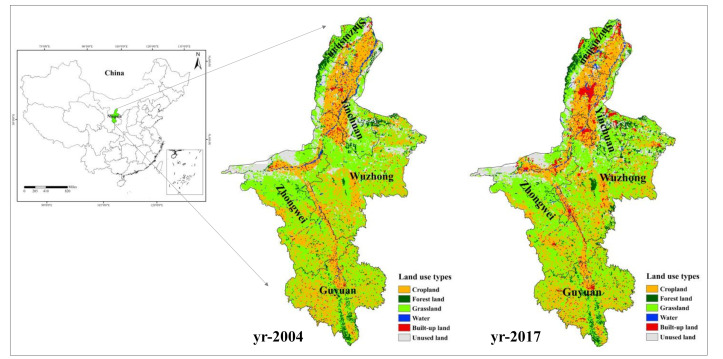
Study area.

**Figure 2 ijerph-18-00646-f002:**
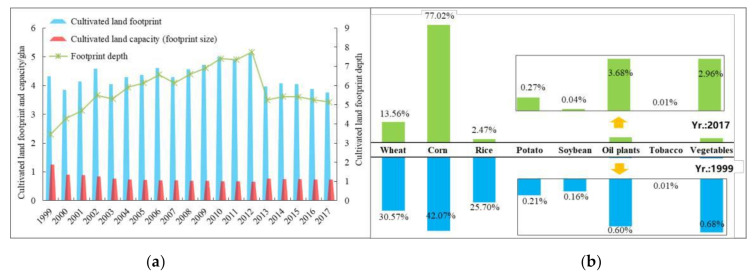
Changes in cultivated land’s natural capital use (**a**) and compositions of cultivated land footprint (**b**) in Ningxia from 1999 to 2017.

**Figure 3 ijerph-18-00646-f003:**
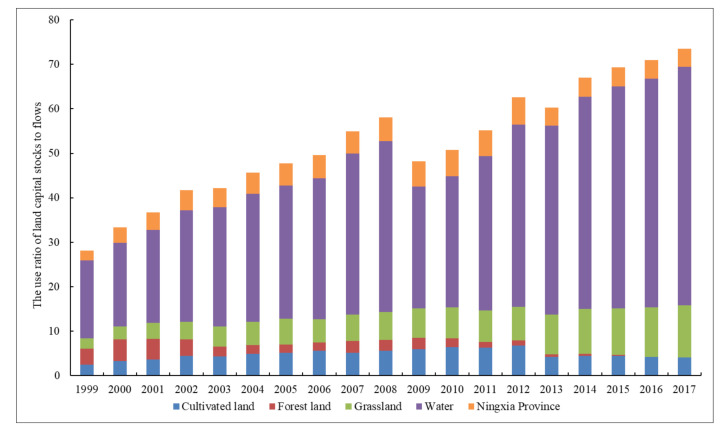
Changes in the $UR_{flo}^{sto}$ of different land use types in Ningxia from 1999 to 2017.

**Figure 4 ijerph-18-00646-f004:**
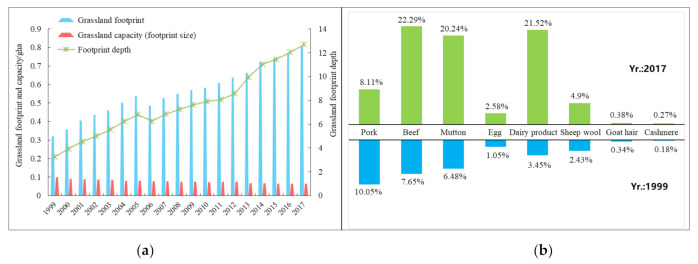
Changes in grassland’s natural capital use (**a**) and compositions of grassland footprint (**b**) in Ningxia from 1999 to 2017.

**Figure 5 ijerph-18-00646-f005:**
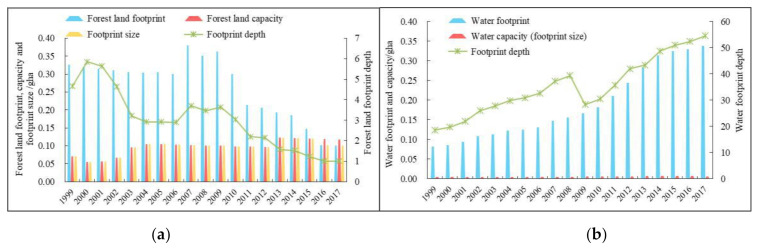
Changes in the natural capital use of forest land (**a**) and water (**b**) in Ningxia from 1999 to 2017.

**Figure 6 ijerph-18-00646-f006:**
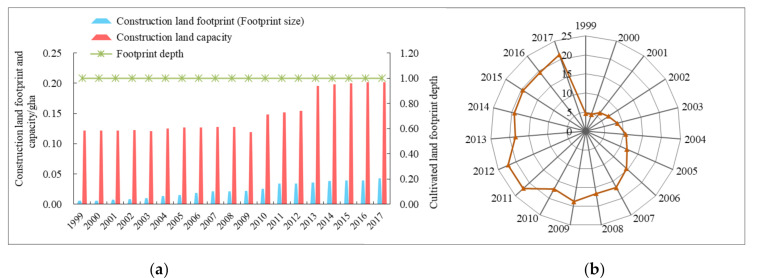
Changes in construction land’s natural capital use (**a**) and utilization efficiency of capital flows (**b**) in Ningxia.

**Figure 7 ijerph-18-00646-f007:**
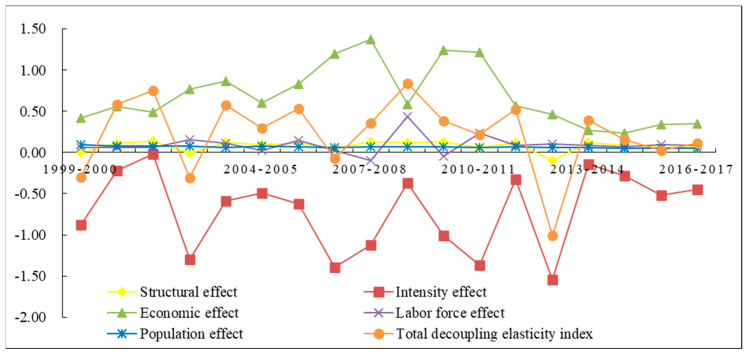
Changes in the total decoupling index and the sub-decoupling index in Ningxia from 1999 to 2017.

**Table 1 ijerph-18-00646-t001:** Six decoupling states.

Decoupling State		$\delta LF_{t}$	$\delta G_{t}$	$\beta_{t}$	Decoupling Type	Meaning
Decoupling	Strong decoupling (SD)	−	+	(−∞,0)	Ⅰ	Economic grows while land natural capital depletion decreases
	Weak decoupling (WD)	+	+	$\left[0,0.8\right)$	Ⅱ	Economic grows while land natural capital depletion decreases slowly
	Recessive decoupling (RD)	−	−	(1.2,+∞)	Ⅲ	Economic grows slowly while land natural capital depletion decreases significantly
Coupling	Expansive coupling (EC)	+	+	$\left[0.8,1.2\right]$	Ⅳ	Economic grows and land natural capital depletion increases moderately
	Recessive coupling (RC)	−	−	$\left[0.8,1.2\right]$	Ⅴ	Economic declines and land natural capital depletion increases moderately
Negative decoupling	Weak negative decoupling (WND)	−	−	$\left[0,0.8\right)\;$	Ⅵ	Economic declines and land natural capital depletion decreases slowly
	Expansive negative decoupling (END)	+	+	$\left[1.2,+\infty \right)\;$	Ⅶ	Economic grows and land natural capital depletion increases significantly
	Strong negative decoupling (SND)	+	−	(−∞,0)	Ⅷ	Economic declines while land natural capital depletion increases

**Table 2 ijerph-18-00646-t002:** Decoupling state between land natural capital utilization and economic growth from 1999 to 2017.

Year	$\delta LF_{t}$	$\delta G_{t}$	$D_{t}$	Decoupling	Year	$\delta LF_{t}$	$\delta G_{t}$	$D_{t}$	Decoupling
1999–2000	−0.088	0.102	−0.860	Ⅰ	2008–2009	0.037	0.119	0.307	Ⅱ
2000–2001	0.076	0.101	0.755	Ⅱ	2009–2010	0.041	0.135	0.305	Ⅱ
2001–2002	0.098	0.102	0.959	Ⅳ	2010–2011	−0.019	0.121	−0.160	Ⅰ
2002–2003	−0.095	0.127	−0.750	Ⅰ	2011–2012	0.043	0.115	0.372	Ⅱ
2003–2004	0.060	0.112	0.532	Ⅱ	2012–2013	−0.172	0.098	−1.754	Ⅰ
2004–2005	0.023	0.109	0.211	Ⅱ	2013–2014	0.036	0.080	0.449	Ⅱ
2005–2006	0.038	0.127	0.299	Ⅱ	2014–2015	−0.008	0.080	−0.104	Ⅰ
2006–2007	−0.033	0.127	−0.263	Ⅰ	2015–2016	−0.032	0.081	−0.391	Ⅰ
2007–2008	0.051	0.126	0.403	Ⅱ	2016–2017	−0.016	0.078	−0.208	Ⅰ

**Table 3 ijerph-18-00646-t003:** The decoupling index of different land capital use and GDP growth.

Year	Cultivated Land	Forest Land	Grassland	Water	Construction Land
1999–2000	−1.081	−0.195	1.121	0.354	−0.027
2000–2001	0.772	−0.156	1.140	0.976	3.167
2001–2002	1.047	−0.143	0.757	1.558	1.733
2002–2003	−0.936	−0.117	0.280	0.358	1.470
2003–2004	0.534	−0.049	0.662	0.739	2.560
2004–2005	0.173	0.030	0.532	0.154	1.149
2005–2006	0.450	−0.115	−0.905	0.338	1.821
2006–2007	−0.553	2.069	0.596	1.034	1.291
2007–2008	0.498	−0.594	0.163	0.427	−0.096
2008–2009	0.293	0.295	0.747	0.640	0.465
2009–2010	0.429	−1.282	0.084	0.642	1.108
2010–2011	−0.157	−2.377	−0.055	1.314	2.577
2011–2012	0.356	−0.328	0.388	1.373	0.098
2012–2013	−2.247	−0.614	0.414	1.639	0.612
2013–2014	0.315	−0.540	0.253	1.368	0.696
2014–2015	−0.114	−2.561	0.530	0.430	0.313
2015–2016	−0.479	−3.795	0.529	0.216	−0.009
2016–2017	−0.417	−0.182	0.423	0.328	1.178

**Table 4 ijerph-18-00646-t004:** The decoupling index of different land capital use and GDP growth.

Year	$\beta_{S}$	$\beta_{I}$	$\beta_{M}$	$\beta_{N}$	$\beta_{P}$	$\beta_{T}$
1999–2000	−0.001	−0.873	0.422	0.055	0.095	−0.301
2000–2001	0.117	−0.220	0.556	0.059	0.075	0.587
2001–2002	0.139	−0.015	0.493	0.062	0.075	0.753
2002–2003	−0.008	−1.299	0.767	0.159	0.076	−0.305
2003–2004	0.118	−0.585	0.863	0.112	0.064	0.572
2004–2005	0.094	−0.496	0.602	0.026	0.074	0.300
2005–2006	0.109	−0.622	0.832	0.148	0.066	0.534
2006–2007	0.049	−1.393	1.194	0.024	0.057	−0.070
2007–2008	0.130	−1.119	1.375	−0.097	0.065	0.353
2008–2009	0.124	−0.366	0.584	0.434	0.067	0.843
2009–2010	0.127	−1.005	1.238	−0.044	0.071	0.387
2010–2011	0.062	−1.364	1.218	0.239	0.060	0.213
2011–2012	0.121	−0.322	0.566	0.084	0.071	0.520
2012–2013	−0.097	−1.541	0.465	0.106	0.059	−1.007
2013–2014	0.118	−0.140	0.274	0.083	0.056	0.391
2014–2015	0.085	−0.280	0.236	0.070	0.048	0.159
2015–2016	0.057	−0.515	0.339	0.094	0.052	0.027
2016–2017	0.072	−0.451	0.351	0.086	0.049	0.108

## Data Availability

No new data were created or analyzed in this study. Data sharing is not applicable to this article.

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
