# Peer review of "Decoupling and Decomposition Analysis of Land Natural Capital Utilization and Economic Growth: A Case Study in Ningxia Hui Autonomous Region, China"

_ijerph, 2021, doi:10.3390/ijerph18020646_

Round 1
Reviewer 1 Report
Review of
Decoupling and Decomposition Analysis of Land Natural Capital Utilization and
Economic Growth in Ningxia, China
I was very impressed by the content of this paper. The authors' excellent empirical test confirmed a broadly held and often repeated basic tenant of environmental sustainability.
At this stage in its development, the paper's presentation quality fails to match its content quality. If this paper is to receive the recognition that I believe it deserves, the authors need to undertake a serious revision. I suggest the following actions:
- The paper needs to be rewritten to improve its communication quality. To guarantee that the rewrite will exhibit the proper use of the English language, I suggest that the authors add a co-author who is both a native English speaker and a scientific scholar who appreciates and understands their work. The rewrite needs to be very thorough. Notice, for example, that the title needs to be revised to improve your message.
- I recommend that you try to reposition the paper based on its theme alone. Ningxia, China should be understood by the reader to be one of several data sources on which you could have relied. Land, natural capital utilization, and economic growth are important topics of this paper. While Ningxia, China will certainly be mentioned, it will not be allowed to be the paper's focal point.
- Your listing of author contributions (lines 476-477) was undoubtedly undertaken with good intention. However, by scholarly tradition, it is off-putting and suggestive of concerns about the authorship team, including the level of collaboration you had. If you look at publications you respect, I doubt that you will see any other similar listing of author contributions. Simply eliminating these two lines removes the possibility that readers may misunderstand your intentions.
- While you certainly recognize that the gap between the years of data you had available and your article's publication date will be large in some spots, you failed to explain this unavoidable problem to your readers and assuage their concerns have about its relevance. This problem can be handled in one short paragraph.
- In several locations, especially in your literature review, your presentation is littered with unnecessary reference numbers. This simple style problem can be eliminated by combining several references in a single endnote. For example, beginning on line 71, you have a sentence that shows 10 separate endnote notations. My recommendation would be to have your rewrite specialist combine these notations into one endnote to improve the paper's readability and appearance.
- If you intend to have your paper referenced by policymakers in the public domain, you need to write to them using language that they can digest and adopt. I suggest that you add a short section at the beginning of the paper and a long section to your finale directed to this nonacademic constituency. While retaining your message's integrity, explain your study and its consequences in language that can be correctly interpreted by policymakers and shared effectively with their audiences.
Consider the final paragraph of the paper (page 14, lines 465-475). Although it is an accurate presentation of your conclusions, the recommendations they suggest are uninspired. I worry that knowledgeable readers might think that these conclusions were to be expected. Worse, the same readers might notice that your findings do not prompt a call for specific responses. If you agree with my concerns, you realize that you are missing a tremendous opportunity to impact your field and influence society. The solution is for you to write boldly about your ideas benefit from your research insights.
I hope that my observations are useful in your development of this fine manuscript. Good luck.
Author Response
Dear reviewer,
Thank you for the comments on our manuscript. We have checked the manuscript and revised it according to the comments. All changes to the original manuscript have been highlighted by using the "Track Changes" function in the revised manuscript. We are grateful to the reviewer for giving this suggestion and please see the attachment.
Thank you again for your time and attention!
Best regards,
Shanshan Guo
Reviewer 2 Report
In general, the paper is well-structured and presents valuable research contents. I recommend that the manuscript can be published, provided that (1) the use of language has been polished, (2) the selection of Ningxia as the case study requires to be justified, and (3) regional comparison can be supplemented.
Author Response
Dear reviewer,
Thank you for the comments on our manuscript. We have checked the manuscript and revised it according to the comments. All changes to the original manuscript have been highlighted by using the "Track Changes" function in the revised manuscript. We are grateful to the reviewer for giving this suggestion and please see the attachment.
Thank you again for your time and attention!
Best regards,
Shanshan Guo
Dear reviewer,
Thank you for the comments on our manuscript. We have checked the manuscript and revised it according to the comments. All changes to the original manuscript have been highlighted by using the "Track Changes" function in the revised manuscript. The detailed responses are listed below. We are grateful to the reviewer for giving this suggestion.
Thank you again for your time and attention!
Best regards,
Shanshan Guo
Point 1: The use of language has been polished.
Response 1: Thanks for the reviewer's kind advice. We have carefully examined the grammar and corrected the improper expression throughout the manuscript. Finally, we invited native speaker to improve the overall language of our manuscript.
Point 2: The selection of Ningxia as the case study requires to be justified.
Response 2: Thanks to the reviewer for giving this suggestion. In the section of “Introduction”, we have added the typicality of selecting Ningxia as the research area in Lines 62-77.
Lines 62-77: Ningxia Hui Autonomous Region (Ningxia) is located primarily in arid and semi-arid zones, belonging to the typical ecologically fragile region and one of the most desertified provinces in China. Beset by drought and lack of water resources as well as unreasonable resource utilization and long-term overgrazing and overcultivation, the fragile ecological environment is facing greater challenges, seriously restricting the sustainable development of society and economy in Ningxia. In order to better develop the pilot area of ecological civilization construction and industrial transformation in western region of China, National Development and Reform Commission (NDRC) of Ningxia issued the Comprehensive Management Plan for Key Ecological Areas in Western Region (2012-2020) and Action Plan of Boosting the Development and Transition of the Western Region (2013-2020), highlighting that Ningxia should undertake important task of ecological management and ecological construction in the western region of China. Therefore, as a typical representative of the western provinces in both fragile environment and economic transformation, research on the characteristics of land natural capital utilization (LNCU), decoupling economic outputs from land occupancy, as well as the primary factors affecting their decoupling nexus, is conducive to achieving the goal of accelerating natural capital appreciation meanwhile providing guidance to the economic transition in ecologically fragile regions.
Point 3: Regional comparison can be supplemented.
Response 3: Thanks for the reviewer's kind advice. We have added the comparison with previous studies when applied the improved method in the manuscript (Lines 316-319). Apart from that, we compared the results of our study with relevant studies (Lines 326-328). In view of the limited research on the decoupling analysis between land natural capital utilization and economic growth, there are not many references we can compare.
Lines 295-298: Based on the model hypothesis, the land capital flow is assumed to be used first, followed by capital stock[8]. However, during the process of using actual land natural capital, the footprint size (capacity) of grassland was rather low and had fallen by 56.28% since 1999, which exacerbated the contradiction between supply and demand of grassland resources.
Lines 316-319: It is noted that if the energy footprint was incorporated into the forestland to calculate the footprint depth according to the practice of Niccolucci[19], the forestland footprint depth of Ningxia in 2017 was up to 12.69[60], which is obviously too pessimistic to reflect the good state of the local forest ecosystem.
Lines 326-330: Then, the water footprint depth soared again from 28.337 in 2009 to 54.578 in 2017 after a brief decline in 2009, much higher than the water footprint in eastern China (2.67 in Zhejiang Province)[61], suggesting that water is the most unsustainable development sector among all land types and almost 55 times the current area is needed to sustain the water resource consumption.
Reviewer 3 Report
This paper analyzes a topic that is very important today and its publication could be interesting. However, there are some aspects that the authors should improve.
1. In relation to the introduction, it is very short. It would be important to expand the introduction highlighting the objective, the contributions, and the importance of this research on the environmental issue. Why the Ningxia region?
2. The paper is well structured but could improve the clarity of the results presented.
3. It is important to argue the methods used. Why these ones and not others? what are its advantages and disadvantages?
4. Authors should expand the implications of their conclusions. In addition, they should comment on the limitations of the study and future lines of research on the subject.
Author Response
Dear reviewer,
Thank you for the comments on our manuscript. We have checked the manuscript and revised it according to the comments. All changes to the original manuscript have been highlighted by using the "Track Changes" function in the revised manuscript. We are grateful to the reviewer for giving this suggestion and please see the attachment.
Thank you again for your time and attention!
Best regards,
Shanshan Guo
Dear reviewer,
Thank you for the comments on our manuscript. We have checked the manuscript and revised it according to the comments. All changes to the original manuscript have been highlighted by using the "Track Changes" function in the revised manuscript. The detailed responses are listed below. We are grateful to the reviewer for giving this suggestion.
Thank you again for your time and attention!
Best regards,
Shanshan Guo
Point 1: In relation to the introduction, it is very short. It would be important to expand the introduction highlighting the objective, the contributions, and the importance of this research on the environmental issue. Why the Ningxia region?
Response 1: Many thanks for the reviewer's suggestion. We have expanded the “Introduction” section, and supplemented the objective and the contributions in Lines 57-61 and Lines 152-160. Additionally, according to the opinions of the reviewer, we also added the typicality of selecting Ningxia as the research area and the importance of this research on the environmental issue in Lines 62-77.
Lines 57-61: This highlights the urgency for well-formulated metrics that could help policymakers better understand the progress on resource sustainable use regionally. But on the whole, there were few studies on qualify the relationship between economic development with ecological quality and environmental quality at home and abroad, and lacked of relevant researches on the utilization of resources in different stages of economic development.
Lines 62-77: Ningxia Hui Autonomous Region (Ningxia) is located primarily in arid and semi-arid zones, belonging to the typical ecologically fragile region and one of the most desertified provinces in China. Beset by drought and lack of water resources as well as unreasonable resource utilization and long-term overgrazing and overcultivation, the fragile ecological environment is facing greater challenges, seriously restricting the sustainable development of society and economy in Ningxia. In order to better develop the pilot area of ecological civilization construction and industrial transformation in western region of China, National Development and Reform Commission (NDRC) of Ningxia issued the Comprehensive Management Plan for Key Ecological Areas in Western Region (2012-2020) and Action Plan of Boosting the Development and Transition of the Western Region (2013-2020), highlighting that Ningxia should undertake important task of ecological management and ecological construction in the western region of China. Therefore, as a typical representative of the western provinces in both fragile environment and economic transformation, research on the characteristics of land natural capital utilization (LNCU), decoupling economic outputs from land occupancy, as well as the primary factors affecting their decoupling nexus, is conducive to achieving the goal of accelerating natural capital appreciation meanwhile providing guidance to the economic transition in ecologically fragile regions.
Lines 152-160: The preceding literatures lay important theoretical foundations and provide feasible and effective methods to study the situation of natural capital utilization as well as the decoupling relationship between resource depletion and economic growth. However, there is little research on the LNCU and its nexus with economic growth. The contributions of this article are focused on the following three aspects: (1)The status of LNCU is analyzed based on the improved EF method in Ningxia during 1999-2017; (2) The Tapio model is introduced to examine the decoupling nexus between LNCU and economic growth; (3) Instead of analyzing the influencing factors of LNCU, this paper systematically studies the major factors affecting the decoupling relationship with combination of the Logarithmic Mean Divisia Index (LMDI) method and Kaya identity equation.
Point 2: The paper is well structured but could improve the clarity of the results presented.
Response 2: Thanks for the reviewer's kind advice. We have carefully checked the “Results and discussion” section, and corrected the incorrect expressions in order that readers can have a better understand of our study.
Point 3: It is important to argue the methods used. Why these ones and not others? what are its advantages and disadvantages?
Response 3: Many thanks for the reviewer's suggestion. We have adjusted the order of the content and added advantages and disadvantages of the methods used in the paper, listing in Lines 81-86, 95-107, and 135-144.
Lines 81-86: The Ecological Footprint (EF) is defined as a bio-productive land area that maintains human living needs while absorbing pollution caused by human activities from the ecological perspective, which has been broadly used to reflect environmental degradation because it focuses on production and consumption activities on the environment both directly and indirectly. The outstanding merits of this method are that it is easy to apply, repeatable, and simple to understand.
Lines 95-107: While this metric is appealing as a communication tool for showing human impact on the natural resources and environment, its methodology and usefulness have also been challenged. Previous researches mostly involve the ‘‘real state” ecological footprint (land footprint) and ‘‘virtual state” ecological footprint (energy footprint) into natural capital accounting for the purpose of incorporation of the key natural capitals that underpin human society as fully as possible. But the point is that energy footprints do not exist actual ecological capacity correspondingly, which results in an irrationally high of land footprint depth. Therefore, Fang discussed defining the “real state” land ecological footprint from the perspective of production to account for the dynamic changes of natural capital utilization of urban and rural land in Guiyang city based on the improved three-dimensional ecological footprint. Above all, despite intense research based on the three-dimensional ecological footprint, there are less researches on the LNCU and its balanced relationship with regional economic in terms of ‘‘real state” land footprints.
Lines 135-144: Decoupling indicator method alone cannot evaluate the effects of environmental externalities and obtain the genuine feedback for improvement. To overcome this shortcoming, another group of scholars have concentrated their efforts on investigating the inner mechanism of decoupling by integrating decoupling index with the decomposition approach. At present, the Structural Decomposition Analysis (SDA), the Index Decomposition Analysis (IDA), and the Production-theoretical Decomposition Analysis (PDA) are the three primary types of decomposition approaches. Among which, the Logarithmic Mean Division Index method (LMDI), as the most representative of the IDA decomposition method, performs best because of the advantage of decomposition without residuals. Therefore, it is regarded as the most accurate and practical method in the current decomposition system.
Point 4: Authors should expand the implications of their conclusions. In addition, they should comment on the limitations of the study and future lines of research on the subject.
Response 4: Thanks a lot for the professional advice of the reviewer. We have added the limitations of our study and future lines of research on the subject in Lines 461-476. Apart from that, we expanded the implications of conclusions, making the policy implication put forward in this paper more targeted and applicable (Lines 496-516).
Lines 461-476: Natural capital utilization in typical ecologically sensitive and fragile areas has attracted wide attention. Although existing researches have contributed theoretical foundations and various methods to study the dynamic change of natural capital utilization from multiple scales, the ‘‘real state” land footprint analysis of LNCU research is still insufficient and there are less studies concentrated on the decoupling relationship between natural capital depletion and economic growth. Therefore, in this paper, we established the improved EF model to analysis the states of LNCU in Ningxia Province, introduced Tapio decoupling theory was to explore the nexus between LNCU and economic growth, and decomposed the major factors affecting this relationship in combination with the Kaya identity and LMDI model, which further expands the existing research. However, despite our promising results, there are still some limitations in this study: first of all, we mainly focused on the perspective of production to study the changes in the use of land natural capital, which may produce different results from the perspective of consumption. Second, our study adopts a top-down approach, and most data used are from the Ningxia Statistical Yearbook and the China Statistical Yearbook. When calculating the ecological footprint of the city and county scale, the accuracy of the data is insufficient, and therefore the further research on a small scale should be carried out in combination with field investigation. The primary conclusions are as follows.
Lines 496-516: Given the above findings and regional characteristics of Ningxia, we propose the following policies to promote the rational use of land natural capital and the decoupling of land capital occupation from economic growth in Ningxia based on the partition perspective.
(1) In the central and southern mountains with fragile ecological environment (Guyuan city, South of Wuzhong City and Zhongwei city), considering the reality that grassland resource is abundant but overutilized seriously, measures related to the grassland and forest land protection, such as the Grain for Green Project, the Region-Wide Grazing Ban and the Three North Shelterbelt Forest Program, should be strictly implemented to improve ecological carrying capacity. Apart from that, due to the rapid expansion of urban construction land, later monitoring and management are crucial for the maintenance of ecological restoration achievements.
(2) In the northern Yellow River irrigation area where is featured with higher economic development and better ecological environment (Yinchuan city and Shizuishan city), it is imperative to promote the transformation and upgrading of traditional industries and advance scientific and technological innovation for the purpose of reducing the impediment of economic effect on the decoupling relationship, and improving the intensity effect by improving the efficiency of resource utilization. At the same time, reasonably controlling the boundary of urban space expansion and optimizing the population structure to give full play to the powerful driving effect of talents on the economy instead of the overuse of natural resources.
(2) Moreover, in Ningxia Plain along the Yellow river, it is suggested to vigorously develop ecological, water-saving agriculture, reduce the use of fertilizer and control the loss of water resources, relieving the pressure of extreme shortage of water resources.
Round 2
Reviewer 3 Report
I think the paper has improved enough and could be published.